# Functionalization of Internal Resonance in Magnetically Coupled Resonators for Highly Efficient and Wideband Hybrid Vibration Energy Harvesting

**DOI:** 10.3390/s22197657

**Published:** 2022-10-09

**Authors:** Kaouthar Aouali, Najib Kacem, Noureddine Bouhaddi

**Affiliations:** Department of Applied Mechanics, FEMTO-ST Institute, CNRS/UFC/ENSMM/ UTBM, University Bourgogne Franche-Comté, 25000 Besançon, France

**Keywords:** vibration energy harvesting, coupled oscillators, hybrid transduction, nonlinear dynamics, internal resonance

## Abstract

The functionalization of internal resonance (IR) is theoretically and experimentally demonstrated on a nonlinear hybrid vibration energy harvester (HVEH) based on piezoelectric (PE) and electromagnetic (EM) transductions. This nonlinear phenomenon is tuned by adjusting the gaps between the moving magnets of the structure, enabling 1:1 and 2:1 IR. The experimental results prove that the activation of 2:1 IR with a realistic excitation amplitude allows the improvement of both the frequency bandwidth (BW) and the harvested power (HP) by 300% and 100%, respectively compared to the case away from IR. These remarkable results open the way towards a very large scale integration of coupled resonators with simultaneous internal resonances.

## 1. Introduction

Over the last few decades, converting ambient vibration energy into electricity to run low-power wireless electronics has received growing interest and continues to grow at a rapid pace [1]. Generally, most vibration harvesters operate at a certain frequency. However, since ambient vibrations distribute in a wide frequency range, the narrow frequency range of the resonators limits their harvesting efficiency when the environment excitation frequency does not match the harvester natural frequency. To avoid this critical issue, several researchers have attempted different methods to extend the frequency bandwidth of the harvesters [2,3]. These strategies include, but are not limited to, linear generators by using a resonance frequency tuning technique and multimodal configurations as examples, introduction of nonlinearities, and advanced electronic networks [4]. Focusing on the mechanical techniques that broaden the energy harvesting, the drawbacks of the mentioned methods are listed. For instance, the resonance frequency tuning approach is based on adjusting the natural frequency of a harvester to the frequency of the source vibrations [5]. To achieve this, complex designs are required and manual or automatic adjustments are needed. This can be done through different techniques, namely preload application [6,7], extensional mode resonator [8,9], adjustment of the geometry [10], and stiffness variation [11]. The disadvantages of this technique are the complexity of the design and ensuring that the amount of power required to tune the frequency is substantially less than the amount being harvested. On the other hand, the multimodal method has been previously proposed [12,13,14,15,16,17]. Several authors developed harvesters with multiple degrees-of-freedom (DOFs) where the power can be scavenged across a wider bandwidth [18,19]. Generally, this configuration is implemented using multiple modes of a bending beam, or by an array of free embedded beams [20,21]. Recent implementations of the concept include multiple different structures where multimodal and multiple-degree-of-freedom harvesters are investigated [22,23,24,25,26]. Although multimodal techniques guarantee the improvement of the harvesters bandwidth, the implementation of electronic networks to harvest the energy from each DOF requires a large volume which leads to technological constraints especially for small-scale harvesters. Add to that the electromechanical complexity of the device and its electrical circuitry results in a higher cost of the device [2,27]. In addition to tuning mechanisms and multimodal configurations, several researchers have developed nonlinear harvesters [28,29,30] which are considered as an alternative approach to overcome the mentioned limitations. Nonlinearities are generally present in dynamic systems because of geometric or material properties [31]. When a nonlinearity is introduced into a system, the frequency response tends to shift to the right (hardening behavior) or to the left (softening behavior) depending on the nonlinearity induced [32,33,34]. It can be introduced in the harvester through different methods, namely, by changing the characteristics of the system [28], imposing high displacements [35], when the resonator interacts with a magnetic field [30,36].

In addition to nonlinear external resonance which results in only softening or hardening nonlinear behavior with the potential for one jump [37,38,39], the internal resonance is a nonlinear phenomenon that occurs in multiple DOF systems and results in a broader bandwidth [40,41,42]. In fact, this phenomenon is characterized by a double-jumping in the frequency response which bends to the left and to the right simultaneously from the central frequency [43]. It occurs when the natural frequencies of the multiple DOFs system are tuned so that they are commensurable, i.e., multiple integers of each others [44,45,46]. To do that, a significant modification in the structure in order to adjust or modify the internal resonance configuration is required. Several authors investigated the internal resonance phenomenon in different ways. For instance, Xu et al. [47] proposed the combination of internal resonance and galloping for a dual-source harvesting. The proposed 2DOF harvester consists of a primary and a secondary beam and presents two energy sources, namely base vibration and wind flow. A significant improvement in the harvester performance has been achieved. Furthermore, Yang et al. [48] studied the combination of internal resonance with bi-stability. They proposed a piezoelectric (PE) cantilever beam which carries a moving magnet facing another fixed one. Similarly, the proposed original combination results in an important enhancement of the energy conversion efficiency. Additionally, other authors investigated the IR phenomenon in devices based on PE transduction [49,50,51,52,53]. In fact, the energy is harvested only from the PE patches while the electromagnetic (EM) component is designed to tune the natural frequencies of the harvesters. For instance, Xiong et al. [49] proposed a device composed of a main beam harvester magnetically coupled to an auxiliary beam to exhibit the nonlinearity and obtain commensurable natural frequencies while the energy is harvested only from the PE element.

In this paper, the internal resonance is functionalized and tuned in magnetically coupled resonators for hybrid PE-EM vibration energy harvesting, which results in a significant enhancement of the frequency bandwidth and the harvested power. The proposed device consists of two moving magnets elastically guided by means of bi-clamped coupled beams where a piezoelectric patch is clamped at the end of each steel beam. In order to tune the internal resonance, the two magnets’ gaps are adjusted without modifying the structure or adding supplementary one for a basis excitation of $0.7\;g\;(g=9.81\;m\cdot s^{-2})$. The functionalization of IR consists of the implementation of the nonlinear magnetic force with a specific design of the fixed and moving magnets, while its tuning is achieved by the adjustment of the gaps. The total energy is harvested from the hybrid piezoelectric-electromagnetic (PE-EM) transductions. Moreover, the results in terms of frequency responses are derived from an analytical multiphysics model, solved using the multiple scales method (MMS), and experimental tests are performed. In particular, it is shown that the functionalization of the 2:1 IR allows a significant improvement of both the frequency bandwidth and the harvested energy over the case away from IR.

## 2. Model and Experimental Setup

The designed hybrid vibration energy harvester (HVEH) integrating simultaneously PE-EM conversion mechanisms is depicted in Figure 1a. It includes two piezoelectric layers attached on the top surface of each beam near the clamped ends. It consists of two center movable coupled magnets supported by the compound elastic beams. The coupling between the magnets is tuned by varying the gaps using the threaded rigid bars. The magnetic poles are oriented in such a way that magnetic repulsive forces are created between each two adjacent magnets. Unlike the lower moving magnet, the upper magnet is not subjected to a magnetic field from a fixed magnet at the top. A wire-wound copper coil is placed around each moving magnet. The characteristics of the used coils, magnets and PE elements are illustrated in Table 1 and Table 2. Applying a harmonic base excitation, the magnets oscillate around their equilibrium positions and the piezoelectric plates are subjected to mechanical stress. Therefore, EM and PE elements generate output powers based on Lorentz’s Law and direct piezoelectric effect, respectively.

In order to obtain the governing equations of the nonlinear harvester, the fourth order partial differential equations of the continuum multiphysics system are derived using the Hamilton principle. Then, a reduced-order model is generated by the Galerkin method, transforming the continuum system into a finite-degree-of-freedom one in terms of generalized coordinates [35]. Considering $x_{i}$ as the vibration amplitude of the magnet *i*
(i=1,2), $V_{p}^{i}$ as the output voltage of the ith PE element and $I_{i}$ as the output current of the ith EM element (Figure 1d), the governing multi-physics equations of the nonlinear 2DOFs system can be written as follows:(1)$$\left\{\begin{array}{c} m{\ddot{x}}_{1}+c_{t}{\dot{x}}_{1}+k_{mec}x_{1}+F_{mg}^{1}-\theta_{p}V_{p}^{1}\left(t\right)=-m\ddot{Y}\\ m{\ddot{x}}_{2}+c_{t}{\dot{x}}_{2}+k_{mec}x_{2}+F_{mg}^{2}-\theta_{p}V_{p}^{2}\left(t\right)=-m\ddot{Y}\\ I_{i}=\frac{\alpha \;}{R_{em}+R_{int}}\;{\dot{x}}_{i},\;\;i=1,2\\ \frac{V_{p}^{i}}{R_{pe}}+C_{p}{\dot{V}}_{p}^{i}=\theta_{p}{\dot{x}}_{i},\;\;i=1,2. \end{array}\right.$$
where *m* is the mass of the identical magnets $M_{1,2}$, $x_{1,2}$ are the generalized coordinates of the DOFs; $c_{t}$ is the damping coefficient; $k_{mec}$ is the linear mechanical stiffness. $F_{mg}^{i}$ refers to the nonlinear magnetic repulsive force between magnets and $\ddot{Y}$ to the acceleration of the basis excitation. $R_{pe}$ and $R_{em}$ stand for the resistances of the PE and EM elements, respectively, and $R_{int}$ is the coil internal resistance. $C_{p}$ is the equivalent capacitance of the PE element; $\theta_{p}$ is the PE element transfer factor, and α is the electromagnetic coefficient.

It is noted that the electric output of the EM element $V_{em}$ (Figure 1d) can be obtained through this equation $V_{em}^{i}=R_{em}\;I_{i}=R_{em}\;\frac{\alpha \;}{R_{em}+R_{int}}\;{\dot{x}}_{i},\;\;i=1,2$, where the displacement is calculated numerically by solving the equations obtained by the multiple scales method detailed in the Appendix A.

The magnetic force applied to each magnet $M_{i}$ can be written as follows: [54]
(2)$$F_{mg}^{i}=\frac{\mu_{0}\;Q_{M_{i}}}{4\pi }\left(\frac{Q_{M_{i-1}}}{{\left(d_{i}+x_{i-1}-x_{i}\right)}^{2}}-\frac{Q_{M_{i+1}}}{{\left(d_{i+1}+x_{i}-x_{i+1}\right)}^{2}}\right);\;i=1,2.$$
where $d_{i}$ is the gap between the magnets $M_{i-1}$ and $M_{i}$, $Q_{M_{i-1}}=Q_{M_{i}}=Q_{M_{i+1}}=Q_{M}$ is the magnetic intensity of the identical magnets, and $\mu_{0}$ refers to the permeability of free space.

By expanding Equation (Equation 2) in Taylor series up to the third order [35], the equations of motion of the system in Equation (Equation 1) can be written in the following form:(3)$$\left\{\begin{array}{c} {\ddot{x}}_{1}+c\;\dot{x_{1}}+\omega_{0}^{2}\;\left[\left(1+\beta_{1}+\beta_{2}\right)x_{1}-\beta_{2}x_{2}\right]+f_{mg,1}^{1}x_{1}^{2}-f_{mg,2}^{1}{\left(x_{2}-x_{1}\right)}^{2}+f_{mg,3}^{1}x_{1}^{3}\\ -f_{mg,4}^{1}{\left(x_{2}-x_{1}\right)}^{3}-\theta V_{p}^{1}\left(t\right)=-\ddot{Y}\\ {\ddot{x}}_{2}+c\;\dot{x_{2}}+\omega_{0}^{2}\;\left[\left(1+\beta_{2}\right)x_{2}-\beta_{2}x_{1}\right]+f_{mg,1}^{2}{\left(x_{2}-x_{1}\right)}^{2}+f_{mg,2}^{2}{\left(x_{2}-x_{1}\right)}^{3}-\theta V_{p}^{2}\left(t\right)=-\ddot{Y} \end{array}\right.\;$$
where $c=\frac{c_{t}}{m}=\frac{c_{m}+c_{e}}{m}$ is the normalized damping, $c_{m}$ and $c_{e}$ are the mechanical and electrical damping coefficients, respectively, $\omega_{0}=\sqrt{\frac{k_{mec}}{m}}$ is the eigenfrequency of the first decoupled DOF; $\beta_{1}=\frac{k_{mg,1}}{k_{mec}}$ and are $\beta_{2}=\frac{k_{mg,2}}{k_{mec}}$ are the coupling coefficient related to the bottom and top magnets, respectively, $k_{mg,1}=\frac{\mu_{0}\;Q_{M}^{2}}{2\;\pi \;d_{1}^{3}}$ and $k_{mg,2}=\frac{\mu_{0}\;Q_{M}^{2}}{2\;\pi \;d_{2}^{3}};$ are the linear magnetic stiffness related to the bottom and top magnets, respectively. $f_{mg,1}^{1}=\frac{3\;\mu_{0}\;Q_{M}^{2}}{4\;\pi \;d_{1}^{4}}$, $f_{mg,2}^{1}=\frac{3\;\mu_{0}\;Q_{M}^{2}}{4\;\pi \;d_{2}^{4}}$,$f_{mg,3}^{1}=\frac{3\;\mu_{0}\;Q_{M}^{2}}{4\;\pi \;d_{1}^{5}}$ and $f_{mg,4}^{1}=\frac{3\;\mu_{0}\;Q_{M}^{2}}{4\;\pi \;d_{2}^{5}}$ are terms derived from the expansion of $F_{mg}^{1}$. $f_{mg,2}^{2}=\frac{\mu_{0}\;Q_{M}^{2}}{4\;\pi \;d_{2}^{5}}$ and $f_{mg,1}^{2}=\frac{3\;\mu_{0}\;Q_{M}^{2}}{4\;\pi \;d_{2}^{4}}$ are terms derived from the expansion of $F_{mg}^{2}$. $\theta =\frac{\theta_{p}}{m}$ is the normalized transfer factor. It is noted that the quadratic terms give rise to the activation of the IR phenomenon, and the cubic ones are responsible of the softening or hardening behaviors according to the signs.

The influence of electrical networks on mechanics and vice versa is studied. The load resistance is inversely proportional to the electrical damping. Therefore, in the neighborhood of the eigenfrequency, each increase of the load resistance will cause the increase of the vibration amplitude. Indeed, the increase of 1%, 3% and 7% in the load resistance (of the PE element for example) results in a 3%, 4% and 8% increase of the amplitude of vibration, respectively. It can be seen that the resistive loads affect the harvester dynamics. Inversely, a change of 1%, 3% and 5% of the second frequency induces a change of 3%, 6% and 9% in the output voltage. This also shows that the mechanical parameter affects the electrical output.

The linear natural frequencies $\omega_{1}$ and $\omega_{2}$ of the 2DOFs harvester for undamped free vibrations are such that
(4)$$\omega_{1,2}=\omega_{0}\frac{\sqrt{2\left(1+\beta_{1}\right)+\beta_{2}\mp \sqrt{4\beta_{1}^{2}+\beta_{2}^{2}}}}{\sqrt{2}}$$

Based on Equation (Equation 4), it is shown that the natural frequencies of the system can be tuned while tuning the linear magnetic force, which is a function of the separation distance between the magnets defined by the gaps $d_{1}$ and $d_{2}$. By tuning the distance between the two magnets, the relationship between the first and second natural frequencies becomes commensurable with different ratios. It has been shown that 1:1 IR, which corresponds to a nonlinear mode-localized configuration [55], is possible for $d_{1}=50$ mm and $d_{2}=50$ mm. Moreover, as shown in Figure 2a, the separation distances $d_{1}$ and $d_{2}$ can be adjusted so that a modal interaction of 2:1 ratio is achieved and thus 2:1 internal resonance occurs. Henceforward, we define $d_{1}^{*}$ and $d_{2}^{*}$ as the well-chosen gaps. Setting $d_{2}^{*}=35\;$ mm, the required distance $d_{1}^{*}$, satisfying that the second modal frequency of the structure is nearly twice its first modal frequency, is equal to 12.74 mm.

## 3. Results and Discussion

In order to investigate the benefits of the internal resonance on the harvester performances, an analytical study was conducted where the method of multiple scales (MMS) [43] is used to solve the coupled differential equations. Some terms of Equation (Equation 3) do not significantly affect the response of the harvester and are, therefore, dropped. The resolution procedure of the equations is developed in the Appendix A. The first primary resonance is described by the following relations $\Omega =\omega_{1}+\varepsilon \;\sigma_{0},\;\omega_{2}=2\;\omega_{1}+\varepsilon \;\sigma_{1}$, where Ω is the excitation frequency, $\sigma_{0}$ and $\sigma_{1}$ are detuning parameters to describe the nearness of Ω to $\omega_{1}$ and $\omega_{2}$ to $2\;\omega_{1}$, respectively and ε is a scaling parameter. Similarly, the second primary resonance is described by the following relations $\Omega =\omega_{2}+\varepsilon \;\sigma_{0},\;\omega_{2}=2\;\omega_{1}+\varepsilon \;\sigma_{1}$.

While fixing the well-chosen values of the gaps $d_{1}^{*}$ and $d_{2}^{*}$, two peaks having almost the same amplitude appear around the first and the second natural frequencies as depicted in Figure 2b,c, respectively. The voltage response curves comparison for the cases with and without the 2:1 frequency tuning is illustrated in Figure 2b. The nonlinear response away from internal resonance (NIR) is obtained for gaps equal to 50 mm. While the frequency response with functionalization of the IR phenomenon bends to the left and to the right around the central frequency, the case without tuning presents a single peak. The existing of two bending branches bending to opposite directions of the central frequency of 92 Hz results in a significant increase in the amplitude and a broadening in the bandwidth. In fact, an enhancement of 35% in voltage amplitude is demonstrated. Moreover, the frequency bandwidth is increased by 140%, which corresponds to a frequency range of 13 Hz.

In order to highlight these effects, experimental tests have been realized. To do that, the HVEH was fabricated, and the experimental test bench was set as shown in Figure 1b,c, respectively. The power of nonlinear energy harvesting is evaluated according to $P=V^{2}/R_{load}$ for different load resistance at the base excitation level of 0.7g, where $R_{load}$ is either $R_{pe}$ or $R_{em}$ according to the conversion mechanism. The piezoelectric patches and the coils are connected separately. Each component is connected to a separate load resistance. Throughout the experiments, the voltage frequency responses are recorded in terms of the root-mean-square (RMS) value. Up and down sweeps are performed during the experiments to capture the bifurcation points of the nonlinear frequency response for a base acceleration of 0.7g. As depicted in Figure 3a,b, it is shown that the PE load resistance does not affect the EM voltage. However, the variation in EM load resistance significantly changes the amplitude of the harvesting system and thus the voltage and the power of the PE element. By varying the load resistances, the EM and PE optimal resistances maximizing the total PE and EM powers are obtained at $R_{em}^{*}=7\;\Omega$ and $R_{pe}^{*}=1.5\;10^{5}\;\Omega$, respectively, as illustrated in Figure 3c,d. The power density of the EM elements for $R_{em}^{*}$ is $12.7\;W\cdot m^{-3}\cdot g^{-2}$ and the one of the PE elements for $R_{pe}^{*}$ is $1.22\;W\cdot m^{-3}\cdot g^{-2}$. The optimal load resistances ($R_{em}^{*}$ and $R_{pe}^{*}$) as well as the well-chosen gap $d_{2}^{*}$ are fixed. For reasons of the experiments tolerance, $d_{1}^{*}$ is fixed to 12.7 mm. Therefore, experimental tests have been performed to determine the nonlinear dynamic behavior of the hybrid harvester. Under these conditions, the second natural frequency $\omega_{2}$ is of 185 Hz and is equal to twice the first natural one $\omega_{1}=92.5$ Hz according to Equation (Equation 4). As shown in Figure 4a, the obtained value of the first natural frequency is experimentally validated. Through experiments, it has been shown that, for $\omega_{2}=175.5$ Hz giving a ratio $\omega_{2}/\omega_{1}=1.9$, the internal resonance phenomenon is activated. In fact, the response curves show the existence of an additional peak that appears around the frequency of the first mode (92.5 Hz) of the bottom DOF where hardening and softening responses are simultaneously observed.

To highlight the importance of the internal resonance phenomenon toward the output performance of the harvester, a configuration away from internal resonance is studied. Based on Equation (Equation 4) and Figure 2a, the distances $d_{1}$ and $d_{2}$ are fixed both to 50 mm such that $\omega_{1}=\omega_{2}$.

Figure 4b shows the output power of the hybrid energy harvester away from the 2:1 internal resonance condition. Only softening nonlinearity is demonstrated. It is observed that the activation of the internal resonance results in increasing both the power density and the frequency bandwidth. An enhancement of 100% of power density is achieved compared to the generator away from the 2:1 internal resonance. Considering this result, it is shown that, for well-chosen gaps resulting in 2:1 internal resonance, it is possible to reduce the volume of the harvester by almost two times by adjusting the gaps and achieve 100% increase in the harvested power. By specifying the internal resonance frequency bandwidth by the half-power bandwidth method in the frequency range that corresponds to the two upward peaks (87 Hz–98 Hz), it is also shown that the internal resonance widens the frequency bandwidth by 300% comparing to the case away from 2:1 internal resonance.

In order to compare the harvester performance to the existing ones, a Systematic Figure of Merit (SFoM) proposed by [56] is used. This SFoM is an effective performance evaluation and design directive tool. It takes into consideration the power density and the frequency bandwidth as follows:(5)$$SFoM=\frac{16\;\pi \;P_{av}\;\Delta f}{a^{2}\;V_{tot}}$$
where $P_{av}$ is the average power, Δf is the frequency bandwidth, *a* is the acceleration (*g* unit), and $V_{tot}$ is the effective volume of the harvester.

According to this figure of merit, the performance of the current work harvester and the ones of other harvesters are illustrated in Figure 5. It can be seen that the proposed harvester provides competitive performance compared to the existing vibration energy harvesters.

The harvested energy from the hybrid structure needs to be stored to power small consumption devices. As the harvester output signals are alternating (AC), they need to be transformed to direct ones (DC) before being stored. Accordingly, an energy storage circuit including the rectifier, the filter, and the signal regulator is designed. The energy acquisition and storage circuit diagram are depicted in Figure 6a. This storage test is a proof of charging. For that, we have chosen, at present, to store the output of one PE element (the right patch clamped to the bottom beam). The outputs are connected to an electronic module to rectify, filter, and then regulate the signals. The practical value of the diodes threshold voltage is of 0.2 V. After being regulated, the electrical harvested voltage is stored in 100μF capacitor. In the experiment, the voltage is recorded by an acquisition card when the harvester is vibrating. As seen in Figure 6b, the maximum voltage obtained is of 0.44 V. The evolution with time on this load capacitor in the charging and discharging phases is illustrated in Figure 6c. The maximum voltage across the capacitor is of 0.38 V. It is shown that the stored battery voltage rises from 0.02 V up to 0.33 V in 1 min under a basis excitation of 0.7 *g*.

The results of this study have to be seen in light of some limitations. The first is the storage of the output energy of the PE and EM elements. In this article, the energy is stored from the output of one PE element. An adaptation of the EM and PE elements’ outputs should be conducted in order to be able to store the two outputs in the same storage device. The second one concerns the miniaturization of the system, which is limited because of the constraints related to magnet downscaling. Consequently, other technologies should be proposed to replace the magnetic coupling, namely the elastic coupling using an elastic beam [65]. The third limitation consists of the type of excitation. In fact, this study is based on the harmonic excitation. However, for a more practically available ambient source, the vibration energy should be harvested when the structure is subjected to random excitation.

## 4. Conclusions

In conclusion, the functionalization and the tuning of the phenomenon of internal resonance are demonstrated in this paper. This results in a significant enhancement of the frequency bandwidth and the harvested power of the nonlinear hybrid piezoelectric-electromagnetic harvester. A prototype of this 2DOF harvester is designed with well-chosen parameters of the PE and EM transductions. These design parameters can be improved by performing a parametric optimization procedure [32,66,67]. By adjusting the gaps between the magnets, the magnetic forces are varied and consequently the natural frequencies of the harvester are tuned so that they are commensurate. Both analytical simulations and experiments capture the nonlinear behavior of the internal resonance phenomenon. It is proved that the activation of the nonlinearity facilitates a nonlinear energy exchange between the commensurate modes. In fact, the existence of two peaks around the first and the second primary resonance results in a wider bandwidth and a larger output. Indeed, it has been demonstrated that, for well-chosen gaps resulting in 2:1 IR, an increase of 100% in the harvested power and an enlargement of 300% in the bandwidth are achieved. For these parameters, the harvester’s volume is reduced by nearly two times compared to the configuration away from IR. A storage test as a proof of charging has been done using a 100 μF capacitor. It has been shown that the voltage raises from 0.02 V to 0.33 V in one minute. This storage test has been realized on the PE element output. Future work will focus on the storage from both the PE and EM elements and will include the extension of the proposed concept to very large scale integrated harvesters based on coupled nonlinear resonators with simultaneous resonances [68].

## Figures and Tables

**Figure 1 sensors-22-07657-f001:**
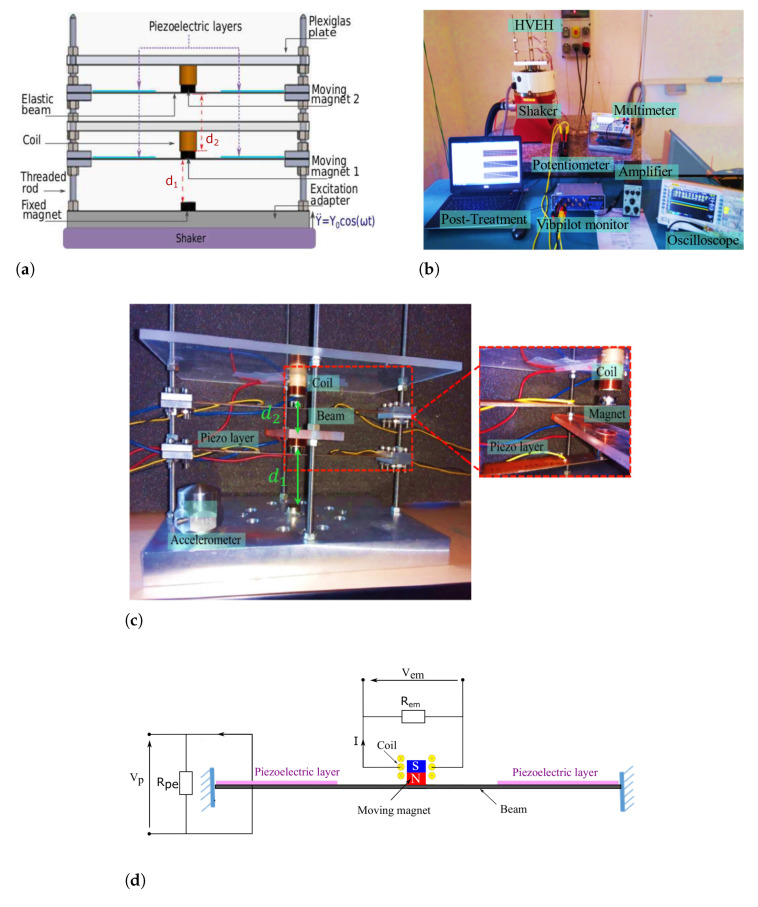
(**a**) The proposed hybrid energy harvester destined to activate the internal resonance phenomenon; (**b**) the experimental setup; (**c**) the proposed hybrid vibration energy harvester (HVEH); (**d**) circuit of the tested piezoelectric $V_{p}$ and electromagnetic voltages $V_{em}$ of the hybrid energy harvesting system.

**Figure 2 sensors-22-07657-f002:**
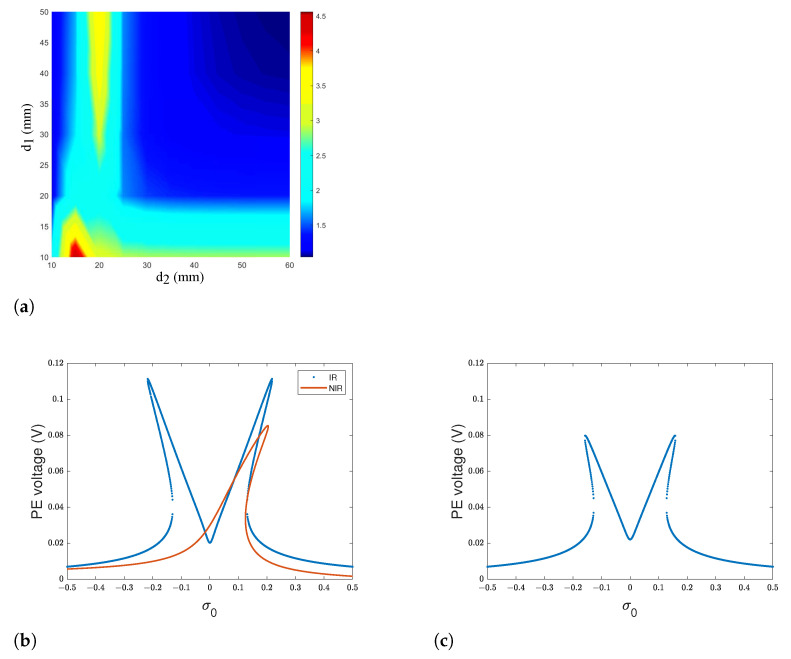
(**a**) Ratio ${}^{\omega_{2}}{/}_{\omega_{1}}$ of the natural frequency $\omega_{2}$ to $\omega_{1}$ while varying the gaps $d_{1}$ and $d_{2}$ (**b**) piezoelectric voltage response curve around the 1st primary resonance for an Internally Resonant (IR) harvester and case of a Non-Internally-Resonant (NIR) harvester; (**c**) voltage response curve around the 2nd primary resonance for IR harvester.

**Figure 3 sensors-22-07657-f003:**
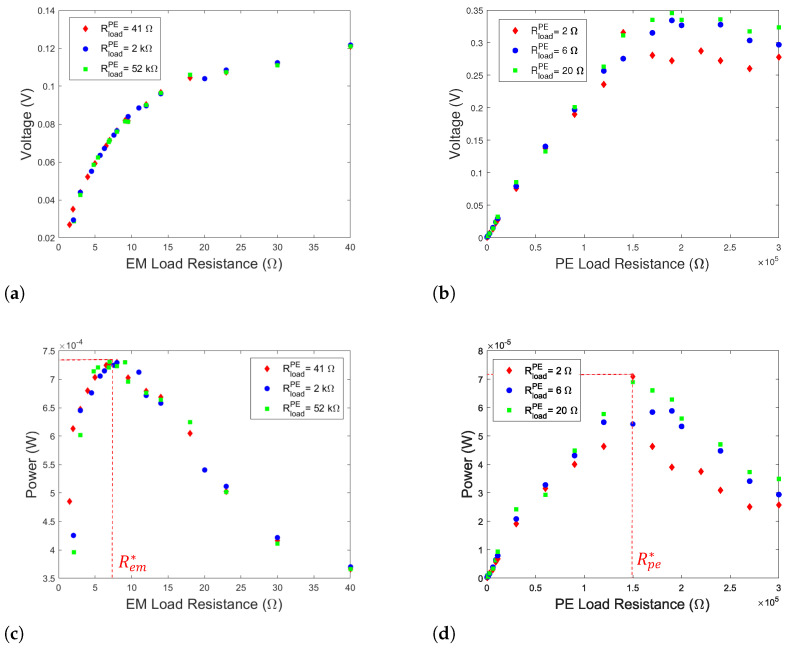
(**a**) Voltage of the bottom EM elements versus the PE load resistances; (**b**) voltage of the PE elements versus the EM load resistances; (**c**) harvested power from the EM elements; (**d**) harvesting power from the PE elements.

**Figure 4 sensors-22-07657-f004:**
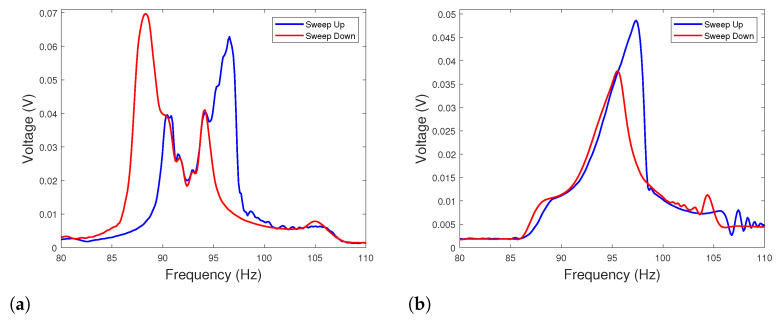
(**a**) Frequency response with optimal parameters ensuring the activation of the internal resonance phenomenon ($d_{1}^{*}=12.74$ mm and $d_{2}^{*}=35$ mm) around the 1st resonance frequency (IR case) $\left(R_{em}^{*}=7\;\Omega /\ddot{Y}=0.7\;g\right)$; (**b**) frequency response away from internal resonance condition ($d_{1}=50$ mm and $d_{2}=50$ mm) around the 1st resonance frequency (NIR case) $\left(R_{em}^{*}=7\;\Omega /\ddot{Y}=0.7\;g\right)$.

**Figure 5 sensors-22-07657-f005:**
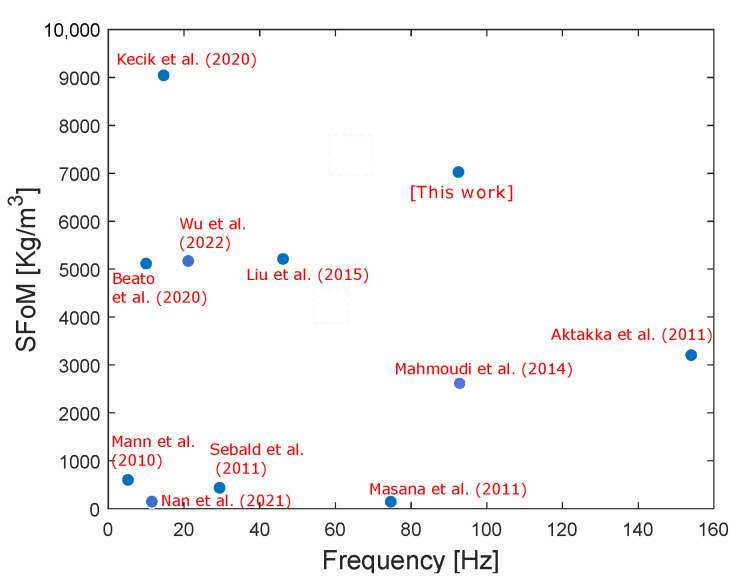
Comparison of the proposed harvester SFoM with the current state-of-the-art [35,56,57,58,59,60,61,62,63,64].

**Figure 6 sensors-22-07657-f006:**
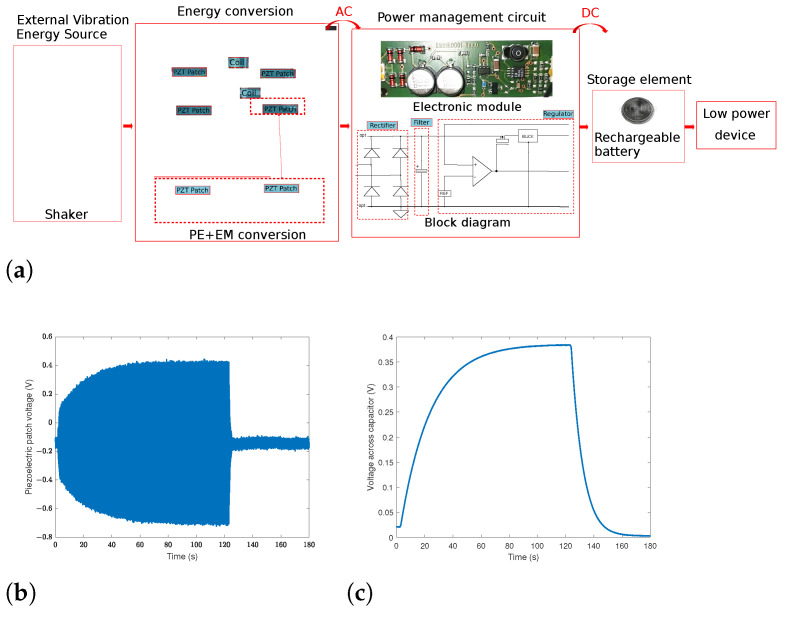
(**a**) Energy acquisition and storage circuit diagram of the HVEH output harvested electrical energy; (**b**) voltage of the piezoelectric patch; (**c**) voltage across a 100 μF capacitor.

**Table 1 sensors-22-07657-t001:** Characteristics of the used coils and magnets.

Parameter	Designation/Value	Unity
Magnet Type	Neodymium magnet	–
Magnetization	N45	—
Magnet diameter	12	mm
Magnet height	4	mm
Residual magnetic field	1.37	T
Type of coil	Copper coil	—
Diameter of coil	14	mm
Number of coil turns *N*	73	—
Internal resistance $R_{int}$	3.4	Ω

**Table 2 sensors-22-07657-t002:** Design parameters of the piezoelectric patches.

Designation	Value	Unity
Piezoelectric coupling coefficient	−13.87	$C\cdot m^{-2}$
Piezoelectric permittivity	$1500\;\epsilon_{0}$	$F\cdot m^{-1}$
Vacuum permittivity $\epsilon_{0}$	$8.854\cdot 10^{-12}$	$F\cdot m^{-1}$
Piezoelectric density	7500	$\mathrm{Kg}\cdot m^{-3}$
Piezoelectric poisson ratio	0.31	—
Piezoelectric layer length	49	mm
Piezoelectric layer width	11	mm
Piezolectric layer Young modulus	69.7	GPa
Thickness of the piezoelectric layer	0.16	mm

## Data Availability

The data that support the findings of this study are available within the article.

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
