# Peer review of "Functionalization of Internal Resonance in Magnetically Coupled Resonators for Highly Efficient and Wideband Hybrid Vibration Energy Harvesting"

_sensors, 2022, doi:10.3390/s22197657_

Round 1
Reviewer 1 Report
This papers presented a hybrid energy harvester (electromagnetic and piezoelectric). This type of concept has been investigated. However, the authors used fours piezoelectric (at two different clamped beam) and two electromagnetic EH. This could be considered as 3DOF system. It is definitely will widen the bandwidth and the power with under an optimum distance and load resistance.
In the manuscript, it is unclear the connection of the four piezoelectric and electromagnetic. Is it connected individually or all of them connected in series?
Reviewer has concern on the Figure 3 results, this is because the results are not tally (or similar) to the Figure 2 which produce a similar trend. In addition, authors should plot the experiment and theoretical results together verify the mathematical modelling. Else the paper is purely experimental investigation.
Lastly, the authors mentioned that the output signal need to be converted to DC with the energy storage. The authors should investigated more details.
The paper is not explain clearly. The reviewer suggest that the paper should focus a significant novelty and explain in depth.
Comments:
1. Lines 119 to 124, it is mentioned that the variation in the load resistance will varies the vibration amplitude. Where the plot is referring? In addition, what do you mean the variation in amplitude? Explain clearly.
2. All the symbols, e.g, d1, d2 should be show clearly in the figure.
3. Figure 5 are comparing different harvester using SFoM. The reference is only until 2020. It should compare latest papers.
4. Figure 6 should be appeared right after the in-text figure.
Author Response
We appreciate the reviewer’s kind comments. Replies to the comments and suggestions have been made as follows:
This papers presented a hybrid energy harvester (electromagnetic and piezoelectric). This type of concept has been investigated. However, the authors used fours piezoelectric (at two different clamped beam) and two electromagnetic EH. This could be considered as 3DOF system. It is definitely will widen the bandwidth and the power with under an optimum distance and load resistance.
- Remark 1:
In the manuscript, it is unclear the connection of the four piezoelectric and electromagnetic. Is it connected individually or all of them connected in series?
- Response:
The piezoelectric patches and the coils are connected separately. Each component is connected to a separate load resistance.
This sentence is added to the text at P6, L165.
- Remark 2:
Reviewer has concern on the Figure 3 results, this is because the results are not tally (or similar) to the Figure 2 which produce a similar trend. In addition, authors should plot the experiment and theoretical results together verify the mathematical modelling. Else the paper is purely experimental investigation.
- Response:
As a first step, the analytical model is conducted to study the nonlinear behavior of the harvester, to tune the natural frequencies which is the necessary condition to achieve the 2:1 internal resonance and to investigate its benefits on the harvester performances. Moreover, the proposed model captures the M-shaped frequency response (large bandwidth) which is confirmed experimentally.
- Remark 3:
The authors mentioned that the output signal need to be converted to DC with the energy storage. The authors should investigated more details.
- Response:
The harvester outputs alternating signals. However, alternating current (AC) cannot be stored directly in a storage device. In fact, because the positive half of AC's cycle charges the battery while the negative half discharges it, storage reservoir is not ultimately charged. For that, AC needs to be transformed to direct ones DC before being stored.
The authors would like to thank the reviewer for the remark, but for the sake of simplifying the text, we think it’s preferable not to include those details.
- Remark 4:
The paper is not explain clearly. The reviewer suggest that the paper should focus a significant novelty and explain in depth
- Response:
For more clarity, the sentence focusing on the novelty of the paper has been modified. (P2, L73)
- Comment 1:
Lines 119 to 124, it is mentioned that the variation in the load resistance will varies the vibration amplitude. Where the plot is referring? In addition, what do you mean the variation in amplitude? Explain clearly.
- Response:
The load resistance is inversely proportional to the electrical damping. Therefore, neighborhood the eigenfrequency, each increase of the load resistance will cause the increase of the vibration amplitude.
This sentence is added to the text P5, L119.
- Comment 2:
All the symbols, e.g, d1, d2 should be show clearly in the figure.
- Response:
The symbols d1 and d2 were already included in Figure 1c. We added them to Figure 1a.
- Comment 3:
Figure 5 are comparing different harvester using SFoM. The reference is only until 2020. It should compare latest papers. eferencing needs to be significantly updated to include recent work. The first seven references are more than 10 years old, there has been much work in the area since 2010. There are many papers that investigate multimodal and multiple-degree-of-freedom harvesters, these should be referenced.
- Response:
Recent references in the field of multimodal and multiple-degree-of-freedom vibration energy harvesting are added to the introduction (Ref 16-19).
Many works do not mention the figure of merit or do not present all the information necessary for the calculation of the SFoM which makes obtaining a large number of references for the SFoM more difficult.
Two papers published in 2021 and 2022 (Ref 66-67) have been added to the SFoM.
- Comment 4:
Figure 6 should be appeared right after the in-text figure.
- Response:
The placement of the figure is changed.
Reviewer 2 Report
Congratulations on this nice paper. Well done.
Author Response
Thank you.
Reviewer 3 Report
The functionalization of the phenomenon of internal resonance to enhance the performance of a nonlinear hybrid piezoelectric-electromagnetic harvester is performed. The theoretical formulation proposed is validated experimentally.
Remarks:
1. Eq. 1 is introduced without an ref. to literature. Is this formula derived by authors from zero.
2. Notation in formula (1) is not not unified, first x dot is used for time derivatives, latter in last formula detailed form of the time derivative is used.
3. Next sentence is generally not correct: „ A prototype of this 2-DOFs harvester is designed with optimal PE and EM transductions“. Actually optimization was not performed, optimization problem was even not formulated. Instead some parameters were varied and best result was selected.
4. The results can be improved further by performing parametric optimization. This can remain for further study, but can be discussed in introduction. In order to provide convergence to global optimum the AI based evolutionary techniques are preferred (in https://doi.org/10.1016/j.compstruct.2010.01.015, hybrid GA is utilized, etc.)
Author Response
We appreciate the reviewer’s kind comments. Replies to the comments and suggestions have been made as follows:
The functionalization of the phenomenon of internal resonance to enhance the performance of a nonlinear hybrid piezoelectric-electromagnetic harvester is performed. The theoretical formulation proposed is validated experimentally.
- Comment 1:
Eq. 1 is introduced without an ref. to literature. Is this formula derived by authors from zero.
- Response:
The authors would like to thank the reviewer for the important remark. A paragraph and a reference are added to the text (P3, L101).
- Comment 2:
Notation in formula (1) is not not unified, first x dot is used for time derivatives, latter in last formula detailed form of the time derivative is used.
- Response:
The remark has been taken into consideration. (Eq.1)
- Comment 3:
Next sentence is generally not correct: „ A prototype of this 2-DOFs harvester is designed with optimal PE and EM transductions“. Actually optimization was not performed, optimization problem was even not formulated. Instead some parameters were varied and best result was selected.
- Response:
This sentence has been modified. (P10, L 227)
- Comment 4:
The results can be improved further by performing parametric optimization. This can remain for further study, but can be discussed in introduction. In order to provide convergence to global optimum the AI based evolutionary techniques are preferred (in https://doi.org/10.1016/j.compstruct.2010.01.015, hybrid GA is utilized, etc.).
- Response:
The authors would like to thank the reviewer for the important reference. A sentence and this reference are added to the conclusion at P10, L228.
Reviewer 4 Report
This paper investigates hybrid PE-EM vibration energy harvesting by considering the functionalization and tuning of internal resonance in magnetically coupled resonators, and the results show that the 2:1 IR case improve both the frequency bandwidth and the harvested power by 300 % and 100 %, respectively compared to the case away from IR. overall, the manuscript is well organized and the results are presented well, and it can be published if the following questions are addressed well.
1. In the abstract, what is the first “H” in the abbreviation of “HVEH”? it could not be well understood by the readers in the current writing style.
2. As the paper gives the voltage and power output of both EM elements and PE elements in Figure 3, the piezoelectric voltage output can be calculated by the governing multi-physics equations of the nonlinear 2-DOFs system, but how do you calculate the electric output of EM elements, please strengthen this part in this paper.
3. Please add a circuit diagram for the hybrid energy harvesting system as the author show in figure 1(c), and mark out where the tested voltage represents Vem and Vpe.
4. The present study's limitations should be added before moving on to the conclusion section.
5. Conclusion needs to be strengthened, and the outcome of the research should be highlighted.
Author Response
We appreciate the reviewer’s kind comments. Replies to the comments and suggestions have been made as follows:
This paper investigates hybrid PE-EM vibration energy harvesting by considering the functionalization and tuning of internal resonance in magnetically coupled resonators, and the results show that the 2:1 IR case improve both the frequency bandwidth and the harvested power by 300 % and 100 %, respectively compared to the case away from IR. overall, the manuscript is well organized and the results are presented well, and it can be published if the following questions are addressed well.
- Comment 1:
In the abstract, what is the first “H” in the abbreviation of “HVEH”? it could not be well understood by the readers in the current writing style.
- Response:
HVEH is the abbreviation of Hybrid Vibration Energy Harvesting. In the abstract, the sentence where HVEH appears the first time is reformulated for more clarity.
- Comment 2:
As the paper gives the voltage and power output of both EM elements and PE elements in Figure 3, the piezoelectric voltage output can be calculated by the governing multi-physics equations of the nonlinear 2-DOFs system, but how do you calculate the electric output of EM elements, please strengthen this part in this paper.
- Response:
It is noted that the electric output of the EM element can be obtained through this equation Viem=RemIi=Remα(dxi/dt)/(Rem+Rint), i=1,2 where the displacement is calculated numerically by solving the equations obtained by the multiple scales method detailed in the appendix.
This information is added to the text at P4, L108.
- Comment 3:
Please add a circuit diagram for the hybrid energy harvesting system as the author show in figure 1(c), and mark out where the tested voltage represents Vem and Vpe.
- Response:
A figure is added to the paper (Figure 1d).
- Comment 4:
The present study's limitations should be added before moving on to the conclusion section.
- Response:
The following paragraph is added before the conclusion (P10, L227):
The results of this study have to be seen in light of some limitations. The first is the storage of the output energy of the PE and EM elements. In this article, the energy is stored from the output of one PE element. An adaptation of the EM and PE elements outputs’ should be conducted in order to be able to store the two outputs in the same storage device. The second one concernsthe miniaturization of the system, which is limited because of the constraints related to magnet downscaling. Consequently, other technologies should be proposed to replace the magnetic coupling, namely the elastic coupling using an elastic beam [66]. The third limitation consists in the type of excitation. In fact, this study is based on the harmonic excitation. However, for a more practically available ambient source, the vibration energy should be harvested when the structure is subjected to random excitation.
- Comment 5:
Conclusion needs to be strengthened, and the outcome of the research should be highlighted.
- Response:
The conclusion has been modified.
Round 2
Reviewer 1 Report
Thank you for the revision. The reviewer is fine with the correction.
Reviewer 4 Report
All the questions have been addressed well, the manuscript can be accepted in present form.